# Neural Logic Machines

**Honghua Dong**[*1], **Jiayuan Mao**[*1], **Tian Lin**[2], **Chong Wang**[3], **Lihong Li**[2], and **Denny Zhou**[2]

[1] ITCS, IIIS, Tsinghua University `{dhh14, mjy14}@mails.tsinghua.edu.cn`
[2] Google Inc. `{tianlin,lihong,dennyzhou}@google.com`
[3] ByteDance Inc. `chong.wang@bytedance.com`

## Abstract

We propose the Neural Logic Machine (NLM), a neural-symbolic architecture for both inductive learning and logic reasoning. NLMs exploit the power of both neural networks—as function approximators, and logic programming—as a symbolic processor for objects with properties, relations, logic connectives, and quantifiers. After being trained on small-scale tasks (such as sorting short arrays), NLMs can recover lifted rules, and generalize to large-scale tasks (such as sorting longer arrays). In our experiments, NLMs achieve perfect generalization in a number of tasks, from relational reasoning tasks on the family tree and general graphs, to decision making tasks including sorting arrays, finding shortest paths, and playing the blocks world. Most of these tasks are hard to accomplish for neural networks or inductive logic programming alone. [1]

## 1 Introduction

Deep learning has achieved great success in various applications such as speech recognition (Hinton et al., 2012), image classification (Krizhevsky et al., 2012; He et al., 2016), machine translation (Sutskever et al., 2014; Bahdanau et al., 2015; Wu et al., 2016; Vaswani et al., 2017), and game playing (Mnih et al., 2015; Silver et al., 2017). Starting from Fodor & Pylyshyn (1988), however, there has been a debate over the problem of systematicity (such as understanding recursive systems) in connectionist models (Fodor & McLaughlin, 1990; Hadley, 1994; Jansen & Watter, 2012).

Logic systems can naturally process symbolic rules in language understanding and reasoning. Inductive logic programming (ILP) (Muggleton, 1991; 1996; Friedman et al., 1999) has been developed for learning logic rules from examples. Roughly speaking, given a collection of positive and negative examples, ILP systems learn a set of rules (with uncertainty) that entails all of the positive examples but none of the negative examples. Combining both symbols and probabilities, many problems arose from high-level cognitive abilities, such as systematicity, can be naturally resolved. However, due to an exponentially large searching space of the compositional rules, it is difficult for ILP to scale beyond small-sized rule sets (Dantsin et al., 2001; Lin et al., 2014; Evans & Grefenstette, 2018).

To make the discussion concrete, let us consider the classic blocks world problem (Nilsson, 1982; Gupta & Nau, 1992). As shown in Figure 1, we are given a set of blocks on the ground. We can move a block $x$ and place it on the top of another block $y$ or the ground, as long as $x$ is *moveable* and $y$ is *placeable*. We call this operation $\text{Move}(x, y)$. A block is said to be *moveable* or *placeable* if there are no other blocks on it. The ground is always *placeable*, implying that we can place all blocks on the ground. Given an initial configuration of blocks world, our goal is to transform it into a target configuration by taking a sequence of $\text{Move}$ operations.

Although the blocks world problem may appear simple at first glance, four major challenges exist in building a learning system to automatically accomplish this task:

1. The learning system should recover a set of lifted rules (*i.e.*, rules that apply to objects uniformly instead of being tied with specific ones) and generalize to blocks worlds which contain more blocks than those encountered during training. To get an intuition on this, we refer the readers who are not familiar with the blocks world domain to the task of learning to sort arrays (e.g.,

---

[*]indicates equal contribution. This work was done when the first two authors were interns at Google.

[1]Project page: https://sites.google.com/view/neural-logic-machines.

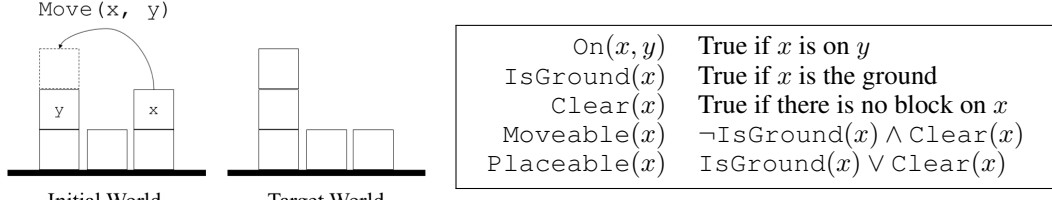

| | |
|---|---|
| $\text{On}(x, y)$ | True if $x$ is on $y$ |
| $\text{IsGround}(x)$ | True if $x$ is the ground |
| $\text{Clear}(x)$ | True if there is no block on $x$ |
| $\text{Moveable}(x)$ | $\neg\text{IsGround}(x) \wedge \text{Clear}(x)$ |
| $\text{Placeable}(x)$ | $\text{IsGround}(x) \vee \text{Clear}(x)$ |

Figure 1: (Left) A graphical illustration of the blocks world. Given an initial and a target worlds, the agent is required to move blocks to transform the initial configuration to the target one. (Right) A set of sentences used throughout the paper to define the blocks world.

    Vinyals et al., 2015), where recurrent neural networks fail to generalize to arrays which are even just slightly longer than those for training.

2. The learning system should deal with high-order relational data and quantifiers, which goes beyond the scope of typical graph-structured neural networks (Kipf & Welling, 2017). For example, to apply the transitivity rule of a relation $r$ , *i.e.* $r(a, c) \leftarrow \exists b \; r(a, b) \wedge r(b, c)$, we need to jointly inspect three objects $(a, b, c)$.

3. The learning system should scale up w.r.t. the complexity of the rules.[2] Existing logic-driven approaches such as traditional ILP methods suffer an exponential computational complexity w.r.t. the number of logic rules to be learned (Dantsin et al., 2001; Lin et al., 2014; Evans & Grefenstette, 2018).

4. The learning system should recover rules based on a minimal set of learning priors. In contrast, traditional ILP methods usually require hand-coded and task-specific rule templates to restrict the size of searching spaces (Evans & Grefenstette, 2018).

In this paper, we propose Neural Logic Machines (NLMs) to address the aforementioned challenges. In a nutshell, NLMs offer a neural-symbolic architecture which realizes Horn clauses (Horn, 1951) in first-order logic (FOL). The key intuition behind NLMs is that logic operations such as logical ANDs and ORs can be efficiently approximated by neural networks, and the wiring among neural modules can realize the logic quantifiers.

The rest of the paper is organized as follows. We first revisit some useful definitions in symbolic logic systems and define our neural implementation of a rule induction system in Section 2. As a supplementary, we refer interested readers to Appendix A for implementation details. In Section 3 we evaluate the effectiveness of NLM on a broad set of tasks ranging from relational reasoning to decision making. We discuss related works in Section 4, and conclude the paper in Section 5.

## 2   Neural Logic Machines (NLM)

The NLM is a neural realization of logic machines (under the Closed-World Assumption[3]). Given a set of base predicates, grounded on a set of objects (the *premises*), NLMs sequentially apply first-order rules to draw *conclusions*, such as a property about an object. For example, in the blocks world, based on premises $\text{IsGround}(u)$ and $\text{Clear}(u)$ of object $u$, NLMs can infer whether $u$ is moveable.

Internally, NLMs use tensors to represent logic predicates. This is done by grounding the predicate as True or False over a fixed set of objects. Based on the tensor representation, rules are implemented as neural operators that can be applied over the premise tensors and generate conclusion tensors. Such neural operators are probabilistic, lifted, and able to handle relational data with various orders (*i.e.*, operating on predicates with different arities).

### 2.1   Logic Predicates as Tensors

We adopt a probabilistic tensor representation for logic predicates. Suppose we have a set of objects $\mathcal{U} = \{u_1, u_2, \ldots, u_m\}$. A predicate $p(x_1, x_2, \ldots, x_r)$, of *arity* $r$, can be grounded on the object set $\mathcal{U}$ (informally, we call it $\mathcal{U}$-*grounding*), resulting in a tensor $p^{\mathcal{U}}$ of shape $[m^{\underline{r}}] \triangleq [m, m-1, m-2, \ldots, m-r+1]$, where the value of each entry $p^{\mathcal{U}}(u_{i_1}, u_{i_2}, \ldots, u_{i_r})$ of the tensor represents whether $p$ is True under the grounding that $x_1 = u_{i_1}, x_2 = u_{i_2}, \cdots, x_r = u_{i_r}$. Here, we restrict that the grounded objects of all $x_i$'s are mutually exclusive, *i.e.*, $i_j \neq i_k$ for all pairs of

---

[2]For a concrete example in the blocks world domain, please refer to Appendix E.
[3]https://en.wikipedia.org/wiki/Closed-world_assumption

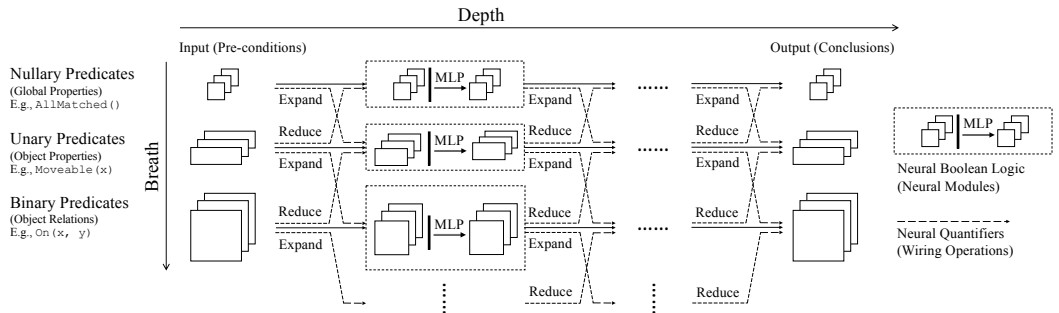

Figure 2: An illustration of Neural Logic Machines (NLM). During forward propagation, NLM takes object properties and relations as input, performs sequential logic deduction, and outputs conclusive properties or relations of the objects. Implementation details can be found in Section 2.3.

indices $j$ and $k$. This restriction does not limit the generality of the representation, as the "missing" entries can be represented by the $\mathcal{U}$-grounding of other predicates with a smaller arity. For example, for a binary predicate $p$, the grounded values of the $p^{\mathcal{U}}(x, x)$ can be represented by the $\mathcal{U}$-grounding of a unary predicate $p'(x) \triangleq p(x, x)$.

We extend this representation to a collection of predicates of the same arity. Let $C^{(r)}$ be the number of predicates of arity $r$. We stack the $\mathcal{U}$-*grounding* tensors of all predicates as a tensor of shape $\left[m^{\underline{r}}, C^{(r)}\right] \triangleq \left[m, m-1, m-2, \ldots, m-r+1, C^{(r)}\right]$, where the last dimension corresponds to the predicates. Intuitively, a group of $C^{(1)}$ unary predicates grounded on $m$ objects can be represented by a tensor of shape $\left[m, C^{(1)}\right]$, describing a group of "properties of objects", while a $\left[m, m-1, C^{(2)}\right]$-shaped tensor for $C^{(2)}$ binary predicates describes a group of "pairwise relations between objects". In practice, we set a maximum arity $B$ for the predicates of interest, called the *breadth* of the NLM.

In addition, NLMs take a probabilistic view of predicates. Each entry in $\mathcal{U}$-*grounding* tensors takes value from $[0, 1]$, which can be interpreted as the probability being `True`. All premises, conclusions, and intermediate results in NLMs are represented by such probabilistic tensors. As a side note, we impose the restriction that all arguments in the predicates can only be variables or objects (*i.e.*, constants) but not function symbols, which follows the setting of *Datalog* (Maier & Warren, 1988).

## 2.2 Logic Rules as Neural Operators

Our goal is to build a neural architecture to learn rules that are both lifted and able to handle relational data with multiple arities. We present different modules of our neural operators by making analogies to a set of essential *meta-rules* in symbolic logic systems. Specifically, we discuss our neural implementation of (1) *boolean logic rules*, as lifted rules containing boolean operations (`AND`, `OR`, `NOT`) over a set of predicates; and (2) *quantifications*, which bridge predicates with different arities by logic quantifiers ($\forall$ and $\exists$).

Next, we combine these neural units to compose NLMs. Figure 2 illustrates the overall multi-layer, multi-group architecture of an NLM. An NLM has layers of *depth* $D$ (horizontally), and each layer has $B + 1$ computation units (vertically). These units operate on the tensor representations of predicates whose arities range from $[0, B]$, respectively. NLMs take input tensors of predicates (premises), perform layer-by-layer computations, and output tensors as conclusions.

As the number of layers increases, higher levels of abstraction can be formed. For example, the output of the first layer may represent `Clear`$(x)$, while a deeper layer may output more complicated predicate like `Moveable`$(x)$. Thus, forward propagation in NLMs can be interpreted as a sequence of rule applications. We further show that NLMs can efficiently realize a partial set of Horn clauses.

We start from the *neural boolean logic rules* and the *neural quantifiers*.

**Boolean logic**. We use the following symbolic meta-rule for boolean logic:

$$\hat{p}(x_1, x_2, \cdots, x_r) \leftarrow \texttt{expression}(x_1, x_2, \cdots, x_r), \tag{1}$$

where `expression` can be any *boolean* expressions consisting of predicates over *all* variables $(x_1, \ldots, x_r)$ and $\hat{p}(\cdot)$ is the conclusive predicate. For example, the rule `Moveable`$(x) \leftarrow \neg$`IsGround`$(x) \wedge$ `Clear`$(x)$ can be instantiated from this meta-rule.

Denote $\mathcal{P} = \{p_1, \ldots, p_k\}$ as the set of $|\mathcal{P}|$ predicates appeared in `expression`. By definition, all $p_i$'s have the same arity $r$ and can be stacked as a tensor of shape $[m^{\underline{r}}, |\mathcal{P}|]$. In Eq. 1, for

a specific grounding of the conclusive predicate $\hat{p}(x_1 \cdots x_r)$, it is conditioned $r! \times |\mathcal{R}|$ grounding values with the same subset of objects, of *arbitrary permutation* as the arguments to all input predicates $\mathcal{P}$. For example, consider a specific ternary predicate $\hat{p}(x_1, x_2, x_3)$. For three different objects $a, b, c \in \mathcal{U}$, the grounding $\hat{p}(a, b, c)$ is conditioned on $p_j(a, b, c), p_j(a, c, b), p_j(b, a, c),$ $p_j(b, c, a), p_j(c, a, b), p_j(c, b, a)$ (all permutations of the parameters) for all $j$ (all input predicates).

Our neural implementation of boolean logic rules is a lifted neural module that uniformly applies to any grounding entries $(x_1 \cdots x_r)$ in the output tensor $\hat{p}^{\mathcal{U}}$. It has a $\mathrm{Permute}(\cdot)$ operation transforming the tensor representation of $\mathcal{P}$, followed by a multi-layer perceptron (MLP). Given the tensor representation of $\mathcal{P}$, for each $p_i^{\mathcal{U}}(x_1, x_2, \ldots, x_r)$, the $\mathrm{Permute}(\cdot)$ operation creates $r!$ new tensors as $p_{i,1}^{\mathcal{U}}, \ldots, p_{i,r!}^{\mathcal{U}}$ by permuting all axes that index objects, with all possible permutations. We stack all to form a $[m^r, r! \times |\mathcal{P}|]$-shaped tensor. An MLP uniformly applies to all $m^r$ object indices:

$$\hat{p}(u_{i_1}, \cdots, u_{i_r}) = \sigma\left(\mathrm{MLP}\left(p_{1,1}(u_{i_1}, \ldots, u_{i_r}), \cdots, p_{k,r!}(u_{i_1}, \ldots, u_{i_r})\right); \theta\right), \quad (2)$$

where $\sigma$ is the sigmoid nonlinearity, $\theta$ is the trainable network parameters. For all sets of mutually exclusive indexes $i_1, \ldots, i_r \in \{1, 2, \ldots, m\}$, the same MLP is applied. Thus, the size of $\theta$ is independent of the number of objects $m$. This property is analogous to the implicit unification property of Horn clauses: the rule $\hat{p}(x) \leftarrow p_1(x) \wedge p_2(x)$ implicitly means, $\forall x\ \hat{p}(x) \leftarrow p_1(x) \wedge p_2(x)$.

**Quantification**. We introduce two types of meta-rules for quantification, namely *expansion* and *reduction*. Let $p$ be a predicate, and we have

**(Expansion)** $\qquad \forall x_{r+1}\ q(x_1, x_2, \cdots, x_r, x_{r+1}) \leftarrow p(x_1, x_2, \cdots, x_r), \qquad (3)$

where $x_{r+1} \notin \{x_i\}_{i=1}^r$. The expansion operation constructs a new predicate $q$ from $p$, by introducing a new variable $x_{r+1}$. For example, consider the following rule

$$\mathrm{ValidMove}(x, y) \leftarrow \mathrm{Moveable}(x) \wedge \mathrm{Placeable}(y).$$

This rule does not fit the meta-rule in Eq. 1 as some predicates on the RHS only take a *subset* of variables as inputs. However, it can be described by using the expansion and the boolean logic meta-rules jointly.

| | |
|---|---|
| 1. $\forall z\ \mathrm{MoveableX}(x, z) \leftarrow \mathrm{Moveable}(x)$ ; | (from Eq. 3) |
| 2. $\forall z\ \mathrm{PlaceableY}(y, z) \leftarrow \mathrm{Placeable}(y)$; | (from Eq. 3) |
| 3. $\mathrm{ValidMove}(x, y) \leftarrow \mathrm{MoveableX}(x, y) \wedge \mathrm{PlaceableY}(y, x).$ | (from Eq. 1) |

The *expansion meta-rule* (Eq. 3) for a set of $C$ $r$-ary predicates, represented by a $[m^r, C]$-shaped tensor, introduces a new and distinct variable $x_{r+1}$. Our neural implementation $\mathrm{Expand}(\cdot)$ repeats each predicate (their tensor representation) for $(m - r)$ times, and stacks in a new dimension. Thus the output shape is $[m^{r+1}, C]$.

The other meta-rule is for reduction:

**(Reduction)** $\qquad q(x_1, x_2, \cdots, x_r) \leftarrow \forall x_{r+1}\ p(x_1, x_2, \cdots, x_r, x_{r+1}), \qquad (4)$

where the $\forall$ quantifier can also be replaced by $\exists$. The reduction operation reduces a variable in a predicate via the quantifier. As an example, the rule to deduce the moveability of objects,

$$\mathrm{Moveable}(x) \leftarrow \neg \mathrm{IsGround}(x) \wedge \neg\left(\exists y\ \mathrm{On}(y, x)\right),$$

can be expressed using meta-rules as follows:

| | |
|---|---|
| 1. $\mathrm{Clear}(x) \leftarrow \forall y\ \neg \mathrm{On}(y, x)$; | (from Eq. 4) |
| 2. $\mathrm{Moveable}(x) \leftarrow \neg \mathrm{IsGround}(x) \wedge \mathrm{Clear}(x).$ | (from Eq. 1) |

The *reduction meta-rule* (Eq. 4) for a set of $C$ $(r+1)$-ary predicates, represented by a $[m^{r+1}, C]$-shaped tensor, eliminates the variable $x_{r+1}$ via quantifiers. For $\exists$ (or $\forall$), our neural implementation $\mathrm{Reduce}(\cdot)$ takes the maximum (or minimum) element along the dimension of $x_{r+1}$, and stacks the two resulting tensors. Therefore, the output shape becomes $[m^r, 2C]$.

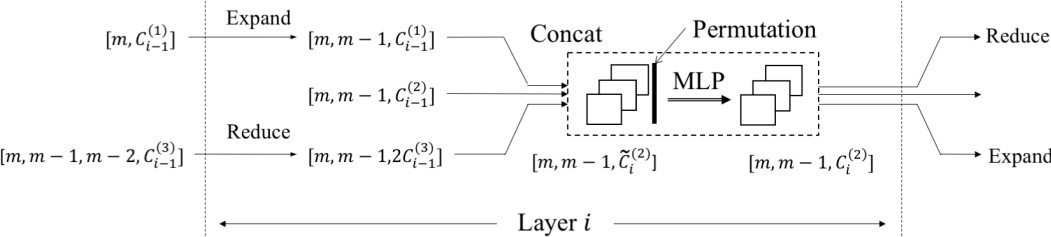

Figure 3: An illustration of the computational block inside NLM for binary predicates at layer $i$. $C_i^{(j)}$ denotes the number of output predicates of group $j$ at layer $i$. $[\cdot]$ denotes the shape of the tensor.

## 2.3 NEURAL LOGIC MACHINES

NLMs realize symbolic logic rules in a multi-layer multi-group architecture, illustrated in Figure 2. An NLM has $D$ layers, and each layer has $B + 1$ computation units as groups. Between layers, we use *intra-group computation* ( Eq. 1). The predicates at each layer are grouped by their arities, and inside each group, we use *inter-group computation* (Eq. 3 and 4).

We define $\mathcal{O}_i = \left\{ O_i^{(0)}, O_i^{(1)}, \cdots, O_i^{(B)} \right\}$ as the outputs of layer $i$, where $O_i^{(r)}$ is the output corresponding to the $r$-ary unit at layer $i$. For convenience, we denote $\mathcal{O}_0 = \left\{ O_0^{(0)}, O_0^{(1)}, \cdots, O_0^{(B)} \right\}$ as the $\mathcal{U}$-grounding tensors for NLM's base predicates (the *premises*), and $\mathcal{O}_D$ at the last layer as the *conclusions*. The overall computation is performed layer-by-layer, from layer 1 to layer $D$. All computation units at layer $i$ work simultaneously, taking $\mathcal{O}_{i-1}$ as inputs and generating $\mathcal{O}_i$.

Let us consider a specific group $r$ at layer $i$, and we show how to calculate $O_i^{(r)}$.

**Inter-group computation.** As shown in Figures 2 and 3, we connect tensors from the previous layer $i - 1$ in vertically neighboring groups (*i.e.* $r - 1$, $r$ and $r + 1$), and aligns their shapes by *expansion* (Eq. 3) or *reduction* (Eq. 4) to form an intermediate tensor $I_i^{(r)}$:

$$I_i^{(r)} = \text{Concat} \left( \text{Expand} \left( O_{i-1}^{(r-1)} \right), O_{i-1}^{(r)}, \text{Reduce} \left( O_{i-1}^{(r+1)} \right) \right). \tag{5}$$

Nonexistent terms are ignored (e.g. when $r + 1 > B$ or $r - 1 < 0$). Note that from the previous layer, $O_{i-1}^{(r-1)}, O_{i-1}^{(r)}, O_{i-1}^{(r+1)}$ have shapes $[m^{\underline{r-1}}, C_{i-1}^{(r-1)}]$, $[m^{\underline{r}}, C_{i-1}^{(r)}]$, $[m^{\underline{r+1}}, C_{i-1}^{(r+1)}]$, respectively. After the concatenation, the resulting tensor $I_i^{(r)}$ is of shape $[m^{\underline{r}}, \widetilde{C}_i^{(r)}]$, where the number of new predicates is $\widetilde{C}_i^{(r)} \triangleq C_{i-1}^{(r-1)} + C_{i-1}^{(r)} + 2C_{i-1}^{(r+1)}$, and the 2 comes from the two quantifiers ($\forall$ and $\exists$).

The inter-group computation essentially aligns predicates of neighboring arities. Relational representations of different orders get combined together through the neural quantification.

**Intra-group computation.** The intra-group computation is implemented as the neural boolean logic in Eq. 1. It take the intermediate tensor $I_i^{(r)}$ as input, permutes and generates the output tensor $O_i^{(r)}$:

$$O_i^{(r)} = \sigma \left( \text{MLP} \left( \text{Permute} \left( I_i^{(r)} \right); \theta_i^{(r)} \right) \right), \tag{6}$$

where $\sigma$ is the sigmoid nonlinearity and $\theta_i^{(r)}$ denotes trainable parameters. We apply $\text{Permute}$ function to $\widetilde{C}_i^{(r)}$ tensors in $I_i^{(r)}$ individually, and get $r!\widetilde{C}_i^{(r)}$ tensors. We set the number of output neurons to be $C_i^{(r)}$, thus the shape of output tensor $O_i^{(r)}$ is $[m^{\underline{r}}, C_i^{(r)}]$.

**Example.** For concreteness, in Figure 3, consider group 2 (*binary* predicates) at layer $i$. The module begins with the inter-group computation. It first collects the output of vertically consecutive groups (unary, binary and ternary) from the previous layer $i - 1$, where their shapes are shown in the figure. Then it uses expansion/reduction to compose the intermediate tensor $I_i^{(2)}$ containing $\widetilde{C}_i^{(2)} \triangleq C_{i-1}^{(1)} + C_{i-1}^{(2)} + 2C_{i-1}^{(3)}$ predicates. For each object pair $(x, y)$, the output $\mathcal{U}$-grounding tensor of predicates is computed by intra-group computation $O_i^{(2)}(x, y) = \text{MLP}(\text{Concat}(I_i^{(2)}(x, y), I_i^{(2)}(y, x)); \theta_i^{(2)})$, and the output shape is $[m, m - 1, C_i^{(2)}]$. The $\text{Concat}(\cdot, \cdot)$ corresponds to the $\texttt{Permute}$ operation, while the MLP is shared among all pairs of objects $(x, y)$.

**Remark.** It can be verified that NLMs can realize the forward chaining of a partial set of Horn clauses. In NLMs, we consider only finite cases. Thus, there should not exist cyclic references of

predicates among rules. The extension to support cyclic references is left as a future work. See the proof in Appendix D. Thus, given the training dataset containing pairs of (*premises*, *conclusions*), NLMs can induce lifted rules that entail the *conclusions* and generalize w.r.t. the number of objects during testing.

## 2.4 EXPRESSIVENESS AND COMPUTATIONAL COMPLEXITY

The expressive power of NLM depends on multiple factors:

1. The *depth D* of NLM (*i.e.*, number of layers) restricts the maximum number of deduction steps.
2. The *breadth B* of NLM (*i.e.*, the maximum number of variables in all predicates considered) limits the arity of relations among objects. Practically, most (intermediate) predicates are binary or ternary and we set $B$ depending on the task (typically 2 or 3, see Table 3 in Appendix B.)
3. The number of output predicates used at each layer ($C_i^{(r)}$ in Figure 3). Let $C = \max_{i,r} C_i^{(r)}$, and this number is often small in our experiments (*e.g.*, 8 or 16).
4. In Eq. 2, the expressive power of MLP (number of hidden layers and number of hidden neurons) restricts the complexity of the boolean logic to be represented. In our experiments, we usually prefer shallow networks (*e.g.*, 0 or 1 hidden layer) with a small number of neurons (*e.g.*, 8 or 16). This can be viewed as a low-dimension regularization on the logic complexity and encourages the learned rule to be simple.

The computational complexity of NLM's forward or backward propagation is $O(m^B DC^2)$ where $m$ is the number of objects. The network has $O(DC^2)$ parameters. Assuming $B$ is a small constant, the computational complexity of NLM is quadratic in the number of allowed predicates.

## 3 EXPERIMENTS

In this section, we show that NLM can solve a broad set of tasks, ranging from relational reasoning to decision making. Furthermore, we show that NLM trained using small-sized instances can generalize to large-sized instances. In the experiments, Softmax-Cross-Entropy loss is used for supervised learning tasks, and REINFORCE (Williams, 1992) is used for reinforcement learning tasks.

Due to space limitation, interested readers are referred to Appendix A for details of training (including curriculum learning) in the decision making tasks, and Appendix B for more implementation details (such as residual connections (He et al., 2016)), hyper-parameters, and model selection criterion.

## 3.1 BASELINES

We consider two baselines as representatives of the connectionist and symbolicist: Memory Networks (MemNN) (Sukhbaatar et al., 2015) and Differentiable Inductive Logic Programming ($\partial$ILP) (Evans & Grefenstette, 2018), a state-of-the-art ILP framework. We also make comparisons with other models such as Differentiable Neural Computer (DNC) (Graves et al., 2016) and graph neural networks (Li et al., 2016) whenever eligible.

For MemNN, in order to handle an arbitrary number of inputs (properties, relations), we adopt the method from Graves et al. (2016). Specifically, each object is assigned with a unique identifier (a binary integer ranging from 0 to 255), as its "name". The memory of MemNN is now a set of "pre-conditions". For unary predicates, the memory slot contains a tuple $(\texttt{id}(x), 0, \texttt{properties}(x))$ for each x, and for binary predicates $p(x, y)$, the memory slot contains a tuple $(\texttt{id}(x), \texttt{id}(y), \texttt{relations}(x, y))$, for each pair of $(x, y)$. Both $\texttt{properties}(x)$ and $\texttt{relations}(x, y)$ are length-$k$ vectors $v$, where $k$ is the number of input predicates. We number each input predicate with an integer $i = 1, 2, \cdots, k$. If object $x$ has a property $p_i(x)$, then $v[i] = 1$; otherwise, $v[i] = 0$. If a pair of objects $(x, y)$ have relation $p_i(x, y)$, then $v[i] = 1$; otherwise, $v[i] = 0$. We extract the key and value for MemNN's to lookup on the given pre-conditions with 2-layer multi-layer perceptrons (MLP). MemNN relies on iterative queries to the memory to perform relational reasoning. Note that MemNN takes a sequential representation of the multi-relational data.

For $\partial$ILP, the grounding of all base predicates is used as the input to the system.

## 3.2 FAMILY TREE REASONING

The family tree is a benchmark for inductive logic programming, where the machine is given a family tree containing $m$ members. The family tree is represented by the following relations (predicates):

Table 1: Comparison among MemNN, $\partial$ILP and the proposed NLM in family tree and graph reasoning, where $m$ is the size of the testing family trees or graphs. Both $\partial$ILP and NLM outperform the neural baseline and achieve perfect accuracy (100%) on test set. Note N/A mark means that $\partial$ILP cannot scale up in 2-OutDegree.

| Family Tree | MemNN | | $\partial$ILP | | NLM (Ours) | |
|---|---|---|---|---|---|---|
| | $m = 20$ | $m = 100$ | $m = 20$ | $m = 100$ | $m = 20$ | $m = 100$ |
| HasFather | 99.9% / 99.9% | 59.8% / 65.2% | 100% | 100% | 100% | 100% |
| HasSister | 86.3% / 85.5% | 59.8% / 66.4% | 100% | 100% | 100% | 100% |
| IsGrandparent | 96.5% / 84.7% | 97.7% / 63.7% | 100% | 100% | 100% | 100% |
| IsUncle | 96.3% / 85.8% | 96.0% / 64.0% | 100% | 100% | 100% | 100% |
| IsMGUncle | 99.7% / 98.4% | 98.4% / 81.7% | 100% | 100% | 100% | 100% |
| Graph | MemNN | | $\partial$ILP | | NLM (Ours) | |
| | $m = 10$ | $m = 50$ | $m = 10$ | $m = 50$ | $m = 10$ | $m = 50$ |
| AdjacentToRed | 95.2% / 94.6% | 93.1% / 91.9% | 100% | 100% | 100% | 100% |
| 4-Connectivity | 92.3% / 90.5% | 81.3% / 88.0% | 100% | 100% | 100% | 100% |
| 6-Connectivity | 67.6% / 58.8% | 43.9% / 67.9% | 100% | 100% | 100% | 100% |
| 1-OutDegree | 99.8% / 99.7% | 78.6% / 81.2% | 100% | 100% | 100% | 100% |
| 2-OutDegree | 81.4% / 61.8% | 96.7% / 87.7% | N/A | N/A | 100% | 100% |

IsSon, IsDaughter, IsFather and IsMother. The goal of the task is to reason out other properties of family members or relations between them. Our results are summarized in Table 1.

For MemNN, we treat the problem of relation prediction as a question answering task. For example, to determine whether member $x$ has a father in the family tree, we input id($x$) to MemNN as the question. MemNN then performs multiple queries to the memory and updates its hidden state. The finishing hidden state is used to classify whether HasFather($x$). For relations (binary predicates), the corresponding MemNN takes the concatenated embedding of id($x$) and id($y$) as the question.

For $\partial$ILP, we take the grounded probability of the "target" predicate as the output; for an NLM with $D$ layers, we take the corresponding group of output predicates at the last layer (for property prediction, we use tensor $O_D^{(1)}$ to represent unary predicates, while for relation prediction we use tensor $O_D^{(2)}$ to represent binary predicates) and classify the property or relation with a linear layer.

All models are trained on instances of size 20 and tested on instances of size 20 and 100 (size is defined as the number of family members). The models are trained with fully supervised learning (labels are available for all objects or pairs of objects). During the testing phase, the accuracy is evaluated (and averaged) on all objects (for properties such as HasFather) or pairs of objects (for relations such as IsUncle). MGUncle is defined as one's maternal great uncle, which is also used by Differentiable Neural Computer (DNC) (Graves et al., 2016). We report the performance of MemNN in the format of Micro / Macro accuracy. We also try our best to replicate the setting used by Graves et al. (2016), and as a comparison, in the task of "finding" the MGUncle instead of "classifying", DNC reaches the accuracy of $81.8\%$.

### 3.3 GENERAL GRAPH REASONING

We further extend the Family tree to general graphs and report the reasoning performance in Table 1.

We treat each node in the graph as an object (symbol). The (undirected) graph is fed into the model in the form of a "HasEdge" relation between nodes (which is an adjacent matrix). Besides, an extra property color represented by one-hot vectors is defined for every node. A node has the property of AdjacentToRed if it is adjacent to a red node by an outgoing edge. $k$-Connectivity is a relation between two nodes in the graph, which is true if two nodes are connected by a path with length at most $k$. A node has property $k$-OutDegree if its out-degree is exactly $k$. The N/A result of $\partial$ILP in the 2-OutDegree task comes from its memory restriction (Evans & Grefenstette, 2018), where 3-ary intentional predicate is required. As an example, a human-written logic rule for 2-OutDegree can be $-\text{OutDegree}(a) \leftarrow \exists_b \exists_c \forall_d \text{HasEdge}(a, b) \wedge \text{HasEdge}(a, c) \wedge \neg\text{HasEdge}(a, d)$ where $a, b, c$ and $d$ are distinct nodes in the graph.

All models are trained on instances of size 10 and tested on instances of size 10 and 50 (size is defined as the number of nodes in the graph).

Table 2: Comparison between MemNN and the proposed NLM in the blocks world, sorting integers, and finding shortest paths, where $m$ is the number of blocks in the blocks world environment or the size of the arrays/graphs in sorting/path environment. Both models are trained on instance size $m \leq 12$ and tested on $m = 10$ or 50. The performance is evaluated by two metrics and separated by "/": the probability of completing the task during the test, and the average `Moves` used by the agents when they complete the task. There is no result for $\partial$ILP since it fails to scale up. MemNN fails to complete the blocks world within the maximum $m \times 4$ `Moves`.

| Task | MemNN | | NLM (Ours) | |
|------|-------|-------|-------|-------|
| | $m = 10$ | $m = 50$ | $m = 10$ | $m = 50$ |
| `BlocksWorld` | 0% / N/A | 0% / N/A | 100% / 12 | 100% / 84 |
| `Sorting` | 100% / 22 | 90% / 986.6 | 100% / 8 | 100% / 45 |
| `Path` | 45% / 13.3 | 12% / 42.7 | 100% / 4 | 100% / 4 |

### 3.4 BLOCKS WORLD

We also test NLM's capability of decision making in the classic blocks world domain (Nilsson, 1982; Gupta & Nau, 1992) by slightly extending the model to fit the formulation of Markov Decision Process (MDP) in reinforcement learning.

Shown in Figure 1, an instance of the blocks world environment contains two worlds: the initial world and the target world, each containing the ground and $m$ blocks. The task is to take actions in the operating world and make its configuration the same as the target world. The agent receives positive rewards only when it accomplishes the task and the sparse reward setting brings significant hardness. Each object (blocks or ground) can be represented by four properties: `world_id`, `object_id`, `coordinate_x`, `coordinate_y`. The ground has a fixed coordinate $(0,0)$. The input is the result of the numeral comparison among all pairs of objects (may come from different worlds). For example, in $x$-coordinate, the comparison produces three relations for each object pair $(i, j), i \neq j$: `Left`$(i, j)$ (whether $i$ is to the left of $j$, or $\mathbf{1}[x_i < x_j]$), `SameX`$(i, j)$ and `Right`$(i, j)$.

The only operation is `Move`$(i, j)$, which moves object $i$ onto the object $j$ in the operating world if $i$ is movable and $j$ is placeable. If the operation is invalid, it will have no effect; otherwise, the action takes effect and the state represented as coordinates will change accordingly. In our setting, an object $i$ is movable iff it is not the ground and there are no blocks on it, *i.e.* $\forall_j \neg(\text{Up}(i, j) \wedge \text{SameX}(i, j))$. Object $i$ is placeable iff it is the ground or there are no blocks on it.

To avoid the ambiguity of the $x$-coordinates while putting blocks onto the ground, we set the $x$ coordinate of block $i$ to be $i$ when it is placed onto the ground. The action space is $(m + 1) \times m$ where $m$ is the number of blocks in the world and $+1$ comes from the "ground". For both MemNN and NLM, we apply a shared MLP on the output relational predicates of each pair of objects $O_D^{(2)}(x, y)$ and compute an action score $s(x, y)$. The probability for `Move`$(x, y)$ is $\propto \exp s(x, y)$ (by taking a `Softmax`). The results are summarized in Table 2. For more discussion on the confidence bounds of the experiments, please refer to Appendix B.6.

### 3.5 GENERAL ALGORITHMS

We further show NLM's ability to excel at algorithmic tasks, such as `Sorting` and `Path`. We view an algorithm as a sequence of primitive actions and cast as a reinforcement learning problem.

**Sorting.** We first consider the problem of sorting integers. Given a length-$m$ array $a$ of integers, the algorithm needs to iterative swap elements to sort the array in ascending order. We treat each slot in the array as an object, and input their index relations (whether $i < j$) and numeral relations (whether $a[i] < a[j]$) to NLM or MemNN. The action space is $m \times (m - 1)$ indicating the pair of integers to be swapped. Table 2 summarizes the learning performance.

As the comparisons between all pairs of elements in the array are given to the agent, sorting the array within the maximum number of swaps is an easy task. A trivial solution is to randomly swap an inversion[4] in the array at each step.

Beyond being able to generalize to arrays of arbitrary length, with different hyper-parameters and random seeds, the learned algorithms can be interpreted as Selection-Sort, Bubble-Sort, *etc*. We include videos demonstrating some learned algorithms in our website.[5]

---

[4]https://en.wikipedia.org/wiki/Inversion_(discrete_mathematics)
[5]https://sites.google.com/view/neural-logic-machines

**Path finding.** We also test the performance of finding a path (single-source single-target path) in a given graph as a sequential decision-making problem in reinforcement learning environment. Given an undirected graph represented by its adjacency matrix as relations, the algorithm needs to find a path from a start node $s$ (with property $\texttt{IsStart}(s) = \texttt{True}$) to the target node $t$ (with property $\texttt{IsTarget}(t) = \texttt{True}$). To restrict the number of deduction steps, we set the maximum distance between $s$ and $t$ to be 5 during the training and set the distance between $s$ and $t$ to be 4 during the testing, which replicates the setting of Graves et al. (2016). Table 2 summarizes the result.

$\texttt{Path}$ task here can be seen as an extension of bAbI task 19 (path finding) (Weston et al., 2015) with symbolic representation. As a comparison with graph neural networks, Li et al. (2016) achieved 99% accuracy on the bAbI task 19. Contrastively, we formulate the shortest path task as a more challenging reinforcement learning (decision-making) task rather than a supervised learning (prediction) task as in Graves et al. (2016). Specifically, the agent iteratively chooses the next node $next$ along the path. At the next step, the starting node will become $next$ (at each step, the agent will move to $next$). As a comparison, in Graves et al. (2016), Differentiable Neural Computer (DNC) finds the shortest path with probability 55.3% in a similar setting.

## 4 RELATED WORKS AND DISCUSSIONS

**ILP and relational reasoning.** Inductive logic programming (ILP) (Muggleton, 1991; 1996; Friedman et al., 1999) is a paradigm for learning logic rules derived from a limited set of rule templates from examples. Being a powerful way of reasoning over discrete symbols, it is successfully applied to various language-related problems, and has been integrated into modern learning frameworks (Kersting et al., 2000; Richardson & Domingos, 2006; Kimmig et al., 2012). Recently, Evans & Grefenstette (2018) introduces a differentiable implementation of ILP which works with connectionist models such as CNNs. Sharing a similar spirit, Rocktäschel & Riedel (2017) introduces an end-to-end differentiable logic proving system for knowledge base (KB) reasoning. A major challenge of these approaches is to scale up to a large number of complex rules. Searching a rule as complex as our $\texttt{ShouldMove}$ example in Appendix E from scratch is beyond the scope of most systems that use weighted symbolic rules generated from templates.

As shown in Section 2.4, both computational complexity and parameter size of the NLM grow *polynomially* w.r.t. the number of allowed predicates (in contrast to the *exponential* dependence in $\partial$ILP (Evans & Grefenstette, 2018)), but factorially w.r.t. the breadth (max arity, same as $\partial$ILP). Therefore, our method can deal with more complex tasks such as the blocks world which requires using a large number of intermediate predicates, while $\partial$ILP fails to search in such a large space.

Our paper also differs from existing approaches on using neural networks to augment symbolic rule induction (Lippi & Frasconi, 2009; Manhaeve et al., 2018). Specifically, we have *no* rule designed by humans as the input or the knowledge base for the model. NLMs are general neural architectures for learning lifted rules from only input-output pairs.

Our work is also related to symbolic relational reasoning, which has a wide application in processing discrete data structures such as knowledge graphs and social graphs (Zhu et al., 2014; Kipf & Welling, 2017; Zeng et al., 2017; Yang et al., 2017). Most symbolic relational reasoning approaches (e.g., Yang et al., 2017; Rocktäschel & Riedel, 2017) are developed for KB reasoning, in which the predicates on both sides of a rule is known in the KB. Otherwise, the complexity grows exponentially in the number of used rules for a conclusion, which is the case in the blocks world. Moreover, Yang et al. (2017) considers rues of the form $\texttt{query}(\texttt{Y}, \texttt{X}) \leftarrow \texttt{R}_\texttt{n}(\texttt{Y}, \texttt{Z}_\texttt{n}) \wedge \cdots \wedge \texttt{R}_\texttt{1}(\texttt{Z}_\texttt{1}, \texttt{X})$, which is not for general reasoning. The key of Rocktäschel & Riedel (2017) and Campero et al. (2018) is to learn subsymbolic embeddings of entities and predicates for efficient KB completion, which differs from our focus. While NLMs can scale up to complex rules, the number of objects/entities or relations should be bounded as a small value (*e.g.*, $< 1000$), since all predicates are represented as tensors. This is, to some extent, in contrast with the systems developed for knowledge base reasoning. We leave the scalability of NLMs to large entity sets as future works.

Besides, modular networks (Andreas et al., 2016; 2017; Mascharka et al., 2018) are proposed for the reasoning over subsymbolic data such as images and natural language question answering. Santoro et al. (2017) implements a visual reasoning system based on "virtual" objects brought by receptive

fields in CNNs. Wu et al. (2017) tackles the problem of deriving structured representation from raw pixel-level inputs. Dai et al. (2018) combines structured visual representation and theorem proving.

**Graph neural networks and relational inductive bias.** Graph convolution networks (GCNs) (Bruna et al., 2014; Li et al., 2016; Defferrard et al., 2016; Kipf & Welling, 2017) is a family of neural architectures working on graphs. As a representative, Gilmer et al. (2017) proposes a message passing modeling for unifying various graph neural networks and graph convolution networks. GCNs achieved great success in tasks with intrinsic relational structures. However, most of the GCNs operate on pre-defined graphs with only nodes and binary connections. This restricts the expressive power of models in general-purpose reasoning tasks (Li et al., 2016).

In contrast, this work removes such restrictions and introduces a neural architecture to capture lifted rules defined on any set of objects. Quantitative results support the effectiveness of the proposed model in a broad set of tasks ranging from relational reasoning to modeling general algorithms (as decision-making process). Moreover, being fully differentiable, NLMs can be plugged into existing convolutional or recurrent neural architectures for logic reasoning.

**Relational decision making.** Logic-driven decision making is also related to Relational RL (Van Otterlo, 2009), which models the environment as a collection of objects and their relations. State transition and policies are both defined over objects and their interactions. Examples include OO-MDP (Diuk et al., 2008; Kansky et al., 2017), symbolic models for learning in interactive domains (Pasula et al., 2007), structured task definition by object-oriented instructions (Denil et al., 2017), and structured policy learning (Garnelo et al., 2016). General planning methods solve these tasks via planning based on rules (Hu & De Giacomo, 2011; Srivastava et al., 2011; Jiménez et al., 2019). The goal of our paper is to introduce a neural architecture which learns lifted rules and handle relational data with multiple orders. We leave its application in other RL and planning tasks as future work.

**Neural abstraction machines and program induction.** Neural Turing Machine (NTM) (Graves et al., 2014; 2016) enables general-purpose neural problem solving such as sorting by introducing an external memory that mimics the execution of Turing Machine. Neural program induction and synthesis (Neelakantan et al., 2016; Reed & De Freitas, 2016; Kaiser & Sutskever, 2016; Parisotto et al., 2017; Devlin et al., 2017; Bunel et al., 2018; Sun et al., 2018) are recently introduced to solve problems by synthesizing computer programs with neural augmentations. Some works tackle the issue of the systematical generalization by introducing extra supervision (Cai et al., 2017). In Chen et al. (2018), more complex programs such as language parsing are studied. However, the neural programming and program induction approaches are usually hard to optimize in an end-to-end manner, and often require strong supervisions (such as ground-truth programs).

## 5 CONCLUSIONS AND DISCUSSIONS

In this paper, we propose a novel neural-symbolic architecture called Neural Logic Machines (NLMs) which can conduct first-order logic deduction. Our model is fully differentiable, and can be trained in an end-to-end fashion. Empirical evaluations show that our method is able to learn the underlying logical rules from small-scale tasks, and generalize to large-scale tasks.

The promising results open the door for several research directions. First, the maximum depth of the NLMs is a hyperparameter to be specified for individual problems. Future works may investigate how to extend the model, so that it can adaptively select the right depth for the problem at hand. Second, it is interesting to extend NLMs to handle vector inputs with real-valued components. Currently, NLM requires symbolic input that may not be easily available in applications like health care where many inputs (*e.g.*, blood pressure) are real numbers. Third, training NLMs remains nontrivial, and techniques like curriculum learning have to be used. It is important to find an effective yet simpler alternative to optimize NLMs. Last but not least, unlike ILP methods that learn a set of rules in an explainable format, the learned rules of NLMs are implicitly encoded as weights of the neural networks. Extracting human-readable rules from NLMs would be a meaningful future direction.

### ACKNOWLEDGEMENTS

We thank Rishabh Singh, Thomas Walsh, the area chair, and anonymous reviewers for their insightful comments.

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

# SUPPLEMENTARY MATERIAL

This supplementary material is organized as follows. First, we provide more details for our training method and introduce the curriculum learning used for reinforcement learning tasks in Appendix A. Second, in Appendix B, we provide more implementation details and hyper-parameters of each task in Section 3. Next, we provide deferred discussion of NLM extensions in Appendix C. Besides, we give a proof of how NLMs could realize the the forward chaining of a set of logic rules defined in Horn cluases. See Appendix D for details. In Appendix E, We also provide a list of sample rules for the blocks world problem in order to exhibit the complexity of describing a strategies or policies. Finally, we also provide a minimal implementation of NLM in TensorFlow for reference at the end of the supplementary material (Appendix F).

## A  TRAINING METHOD AND CURRICULUM LEARNING

In this section, we provide hyper-parameter details of our training method and introduce the exam-guided curriculum learning used for reinforcement learning tasks. We also provide details of the data generation method.

### A.1  TRAINING METHOD

We optimize both NLM and MemNN with Adam (Kingma & Ba (2015)) and use a learning rate of $\alpha = 0.005$.

For all supervised learning tasks (i.e. family tree and general graph tasks), we use Softmax-Cross-Entropy as loss function and a training batch size of $4$.

For reinforcement learning tasks (i.e. the blocks world, sorting and shortest path tasks), we use REINFORCE algorithm (Sutton & Barto (1998)) for optimization. Each training batch is composed of a single episode of play. Similar to A3C (Mnih et al. (2016)), we add policy entropy term in the objective function (proposed by Williams & Peng (1991)) to help exploration. The update function for parameters $\theta$ of policy $\pi$ is

$$\Delta\theta = \alpha[v_t \nabla_\theta \log \pi(a_t|s_t; \theta) + \beta \nabla_\theta H(\pi(s_t; \theta))],$$

where $H$ is the entropy function, $s_t$ and $a_t$ are the state and action at time $t$, $v_t$ is the discounted reward starting from time $t$. The hyper-parameter $\beta$ is set according to different environments and learning stages depending on the demand of exploration.

In all environments, the agent receives a reward of value $1.0$ when it completes the task within a limited number of steps (which is related to the number of objects). To encourage the agent to use as few moves as possible, we give a reward of $-0.01$ for each move. The reward discount factor $\gamma$ is $0.99$ for all tasks.

Table 3: Hyper-parameters for reinforcement learning tasks. The meaning of the hyper-parameters could be found in Section A.1 and Section A.2. For the `Path` environment, the step limit is set to the actual distance between the starting point and the targeting point, to encourage the agents to find the shortest path.

| Task | Range | Step Limit | $\beta_{init}$ | $\Omega$ | Epochs | Train Epoch Episodes | Evaluation Episodes |
|---|---|---|---|---|---|---|---|
| Sorting | $m \in [4, 10]$ | $2m$ | 0.01 | 0.5 | 5 | 200 | 200 |
| Path | $m \in [3, 12]$ | $opt$ | 0.1 | 0.5 | 40 | 600 | 3000 |
| Blocks World | $m \in [2, 12]$ | $4m$ | 0.2 | 0.6 | 50 | 1000 | 3000 |

### A.2  CURRICULUM LEARNING GUIDED BY EXAMS AND FAILS

Inspired by the education system of humans, we employ an exam-guided curriculum learning (Bengio et al., 2009) approach for training Neural Logic Machines. We heuristically label each training

---

**Algorithm 1:** Curriculum learning guided by exams and fails

---

**Function** `train`(*model M, lessons $\mathcal{L}$*):
    **for** $\ell \in \mathcal{L}$ **do**
        **for** $i = 0, 1, \cdots \ell$.`max_epochs` **do**
            $accuracy, pos, neg \leftarrow$ evaluate$(M, \ell)$;        // Take the exam and collect samples.
            **if** $accuracy > \ell$.`threshold` **then**
                **break**;                // Enter the next lesson if pass the exam.
            **for** $j = 0, 1, \cdots K$ **do**
                $data \sim$ balanced sampling from $pos$ and $neg$;
                Optimize $M$ with $data$;

---

instances with its complexity. Training instances are grouped by their complexity (as *lessons*). For example, in the game of BlocksWorld, we consider the number of blocks in a game instance as its complexity. During the training, we present the training instances to the model from lessons with increasing difficulty. We periodically test models' performance (as *exams*) on novel instances of the same complexity as the ones in its current lesson. The well-performed model (whose accuracy reaches a certain threshold) will *pass* the exam and advance to a harder lesson (of more complex training instances). The exam-guided curriculum learning exploits the previously gained knowledge to ease the learning of more complex instances. Moreover, the performance on the final exam reaches above a threshold indicates the *graduation* of models.

In our experiments, each lesson contains training instances of the same number of objects. For example, the first lesson in the blocks world contains all possible instances consisting of 2 blocks (in each world). The instances of the second lesson contain 3 blocks in each world. And in the last lesson (totally 11 lessons) there are 12 blocks in each world. We report the range of the curriculum in Table 3 for three RL tasks.

Another essential ingredient for the efficient training of NLMs is to record models' failure cases. Specifically, we keep track of two sets of training instances: positive and negative (meaning the agent achieves the task or not). For each presented instance of the exam, it is recollected into positive or negative sets depending on whether the agent achieves the task or not. All training samples are sampled from the positive set with probability $\Omega$ and from the negative set with probability $1-\Omega$. This balanced sampling strategy prevents models from getting stuck at sub-optimal solutions. Algorithm 1 illustrates the pseudo-code of the curriculum learning guided by exams and fails.

The evaluation process ("exam") randomly samples examples from 3 recent lessons. The agent goes through these examples and gets the success rate (the ratio of achieving the task) as its performance, which is used to decide whether the agent passes the exam by comparing to a lesson-depend threshold. As we want a perfect model, the threshold for passing the last lesson (the "final exam") is 100%. We linearly decrease the threshold by 0.5% for each former lessons, to prevent over-fitting(*e.g.*, the threshold of the first lesson in the blocks world is 95%). After the "exam", the examples are collected into positive and negative pools according to the outcome (success or not). During the training, we use balanced sampling for choosing training instances from positive and negative pools with probability $\Omega$ from positive. The hyper-parameters $\Omega$, the number of epochs, the number of episodes in each training epoch and the number of episodes in one evaluation are shown in Table 3 for three RL tasks.

## B   Implementation details and hyper-parameters

This section provides more implementation details for the model and experiments, and summarizes the hyper-parameters used in experiments for our NLM and the baseline algorithm MemNN.

### B.1   Residual connection.

Analog to the residual link in (He et al., 2016; Huang et al., 2017), we add residual connections to our model. Specifically, for each layer illustrated in Figure 2, the base predicates (inputs) are

concatenated to the conclusive predicates (outputs) group-wisely. That is, input unary predicates are concatenated to the deduced unary predicates while input binary predicates are concatenated to the conclusive binary predicates.

## B.2 HYPER-PARAMETERS FOR NLM

Table 4 shows hyper-parameters used by NLM for different tasks. For all MLPs inside NLMs, we use no hidden layer, and the hidden dimension (*i.e.*, the number of intermediate predicates) of each layer is set to 8 across all our experiments. In supervised learning tasks, a model is called "graduated" if its training loss is below a threshold depending on the task (usually 1e-6). In reinforcement learning tasks, an agent is called "graduated" if it can pass the final exam, i.e., get 100% success rate on the evaluation process of the last lesson.

We note that in the randomly generated cases, the number of maternal great uncle (`IsMGUncle`) relation is relatively small. This makes the learning of this relation hard and results in a graduation ratio of only 20%. If we increase the maximum number of people in training examples to 30, the graduation ratio will grow to 50%.

Table 4: Hyper-parameters for Neural Logic Machines. The definition of depth and breadth are illustrated in Figure 2. "Res." refers to the use of residual links. "Grad." refers to the ratio of successful graduation in 10 runs with different random seeds, which partially indicates the difficulty of the task. "Num. Examples/Episodes" means the maximum number of examples/episodes used to train the model in supervised learning and reinforcement learning cases.

|  | Tasks | Depth | Breath | Res. | Grad. | Num. Examples/Episodes |
|---|---|---|---|---|---|---|
| Family Tree | `HasFather` | 4 | 3 | × | 100% | 50,000 examples |
|  | `HasSister` | 4 | 3 | × | 100% | 50,000 examples |
|  | `IsGrandparent` | 4 | 3 | × | 100% | 100,000 examples |
|  | `IsUncle` | 4 | 3 | × | 90% | 100,000 examples |
|  | `IsMGUncle` | 4 | 3 | × | 20% | 200,000 examples |
| General Graph | `AdajacentToRed` | 4 | 3 | × | 90% | 100,000 examples |
|  | `4-Connectivity` | 4 | 3 | × | 100% | 50,000 examples |
|  | `6-Connectivity` | 8 | 3 | ✓ | 60% | 50,000 examples |
|  | `1-OutDegree` | 4 | 3 | × | 100% | 50,000 examples |
|  | `2-OutDegree` | 5 | 4 | ✓ | 100% | 100,000 examples |
| General Algorithm | `Sorting` | 3 | 2 | ✓ | 100% | 1,000 episodes |
|  | `Path` | 5 | 3 | ✓ | 60% | 24,000 episodes |
|  | `BlocksWorld` | 7 | 2 | ✓ | 40% | 50,000 episodes |

## B.3 HYPER-PARAMETERS FOR MEMNN

We set the number of iters/episodes used for baseline algorithms to be same as NLM. For the memory networks, each pre-condition in the memory is embedded into a key space and a value space. The dimensions of the spaces are 16 and 32 respectively. The hidden size of the LSTM in MemNN is 64. The number of queries is set to be 4 across all tasks (except that the `Sorting` task uses 1 query only). Empirically, we search for the optimal hyper-parameters but find that they have little effect on the performance.

## B.4 DATA GENERATION

We use random generation to generate training and testing data. more details and specific parameters used to generate the data could be found in our open source code.

In family tree tasks, we mimic the process of families growing using a timeline. For each newly created person, we randomly sample the gender and parents (could be none, indicating not included in the family tree) of the person. We also maintain lists of singles of each gender, and randomly pick two from each list to be married (each time when a person was created). We randomly permute the order of people.

In general graph tasks (include `Path`), We adopt the generation method from Graves et al. (2016), which samples $m$ nodes on a unit square, and the out-degree $k_i$ of each node is sampled. Then each node connects to $k_i$ nearest nodes on the unit square. In undirected graph cases, all generated edges are regarded as undirected edges.

In `Sorting`, we randomly generate permutations to be sorted in ascending order.

In `Blocks World`, We maintain a list of placeable objects (the ground included). Each newly created block places on one randomly selected placeable object. Then we randomly shuffle the $id$ of the blocks.

## B.5 BLOCKS WORLD

In the blocks world environment, to better aid the reinforcement learning process, we train the agent on an auxiliary task, which is to predict the validity or effect of the actions. This task is trained by supervised learning using cross-entropy loss. The overall loss is a summation of cross-entropy loss (with a weight of $0.1$) and the REINFORCE loss.

We did not choose the `Move` to be taken directly based on the relational predicates at the last layer of NLM. Instead, we manually concatenate the object representation from the current and the target configuration, which share the same object ID. Then for each pair of objects, their relational representation is constructed by the concatenation of their own object representation. An extra fully-connected layer is applied to the relational representation, followed by a `Softmax` layer over all pairs of objects. We choose an action based on the Softmax score.

## B.6 ACCURACY DISCUSSION

We cannot directly prove the accuracy of NLM by looking at the induced rules as in traditional ILP systems. Alternatively, we take an empirical way to estimate its accuracy by sampling testing examples. Throughout the experiments section, all accuracy statistics are reported in 1000 random generated data.

To show the confidence of this result, we test a specific trained model of `Blocks World` task with 100,000 samples. We get *no* fail cases in the testing. According to the multiplicative form of Chernoff Bound [6], We are 99.7% confident that the accuracy is at least 99.98%.

## C NEURAL LOGIC MACHINES (NLM) EXTENSIONS

**Reasoning over noisy input: integration with neural perception.** Recall that NLM is fully differentiable. Besides taking logic pre-conditions (binary values) as input, the input properties or relations can be derived from other neural architectures (*e.g.*, CNNs). As a preliminary example, we replace the input properties of nodes with images from the MNIST dataset. A convolutional neural network (CNN) is applied to the input extracting multiple features for future reasoning. CNN and NLM can be optimized jointly. This enables reasoning over noisy input.

We modify the `AdjacentToRed` task in general graph reasoning to `AdjacentToNumber0`. In detail, each node has a visual input from the MNIST dataset indicating its number. We say `AdjacentToNumber0`$(x)$ if and only if a node $x$ is adjacent to another node with number 0. We use LeNet LeCun et al. (1998) to extract visual features for recognizing the number of each node. The output of LeNet for each node is a vector of length 10, with sigmoid activation.

---

[6]`https://en.wikipedia.org/wiki/Chernoff_bound#Multiplicative_form_ (relative_error)`

We follow the train-test split from the original MNIST dataset. The joint model is trained on 100,000 training examples ($m = 10$) and gets 99.4% accuracy on 1000 testing examples ($m = 50$). Note that the LeNet modules are optimized jointly with the reasoning about `AdjacentToNumber0`.

## D    REALIZATION OF HORN CLAUSE

In this section, we show that *NLM can realize a partial set of Horn clauses (Horn, 1951) in first-order logic (FOL), up to the limit of the NLM's depth and breadth.* In NLMs, we consider only finite cases. Thus, there should not exist cyclic references of predicates among rules. The extension to support cyclic references is left as a future work. Throughout the proof, we always assume the depth, breadth and number of predicates of NLM are flexible and large enough to realize the demanding rules.

Here, we only prove the realization of a *definite clause*, *i.e.*, a Horn clause with exactly one positive literal and a non-zero number of negative literals in FOL [7]. It can be written in the implication form is $\hat{p} \leftarrow p_1 \wedge p_2 \wedge \cdots \wedge p_k$ (variables as arguments are implicitly universally quantified), where $\hat{p}$ is called the *head predicate* and $p_1, \ldots, p_k$ are called *body predicates*. We group the variables appearing in the rule into three subsets: (1) variables that only appear in the head predicate, (2) variables that appear in the body predicates, and (3) variables that appear in both head and body predicates.

Consider as an example a chain-like rule: $\forall x_1 \forall x_2 \forall x_3 \forall x_4 \ \hat{p}(x_1, x_3, x_4) \leftarrow p_1(x_1, x_2) \wedge p_2(x_2, x_3)$. We rewrite the rule by classifying the variables:

$$\forall x_4 \Big( \forall x_1 \forall x_3 \ \big( \hat{p}(x_1, x_3, x_4) \leftarrow \exists x_2 \ p_1(x_1, x_2) \wedge p_2(x_2, x_3) \big) \Big) .$$

That is, we move all variables that ony appear in *body predicates* to the right-hand side, and extract out all variables that only appear in the *head predicate*. We show how we can compositionally combines the computation units in NLMs to realize this rule, in the following 4 steps:

1. We first align the arity of the *body predicates* to include all variables that appear in at least one of the *body predicates* (including variables of set 2 and set 3). This could be done by a sequence of `Expand` operations (Eq. 3). In this example, we will create helper predicates to make the right-hand side of the rule as

   $$\exists x_2 \ p_1'(x_1, x_2, x_3) \wedge p_2'(x_2, x_3, x_1),$$

   where $p_1'(x_1, x_2, x_3) \triangleq p_1(x_1, x_2)$ and $p_2'(x_2, x_3, x_1) \triangleq p_2(x_2, x_3)$.

2. We use neural boolean logic (Eq. 1) to realize the boolean formula inside all quantification symbols. Moreover, we use the `Permute` operation to transpose the tensor representation so that all variables being quantified on the right-hand side appear as the last several variables in the derived predicate $p'$. Overall, we will derive another helper predicates,

   $$p'(x_1, x_3, x_2) \triangleq p_1'(x_1, x_2, x_3) \wedge p_2'(x_2, x_3, x_1),$$

3. We use the `Reduce` operation to add quantifiers to the right-hand side (*i.e.*, to the $p'$ predicate). We will get:

   $$p''(x_1, x_3) \triangleq \exists x_2 \ p'(x_1, x_3, x_2) = \exists x_2 \ p_1'(x_1, x_2, x_3) \wedge p_2'(x_2, x_3, x_1),$$

4. Finally, we use the `Expand` operation (Eq. 3] to add variables that only appear in the *head predicate* to the derived predicate:

   $$\hat{p}(x_1, x_3, x_4) \triangleq p''(x_1, x_3).$$

   Note that, all variables appeared in the *head predicate* are implicitly universally quantified. This is consistent with our setting, since all rules in NLMs are lifted.

Overall, a symbolic rule written as a Horn clause can be realized by NLMs as a computation flow which starts from multiple expansions followed by a neural boolean rule and multiple reductions, and ends with a set of expansions.

---

[7]As for other types of Horn clauses: *facts* are realized as the tensor representations of predicates, while we implicitly views the output of NLMs as the *goal clauses*.

Next, we show that the forward propagation of NLMs realizes the forward chaining of a set of Horn clauses. Following the notation in Evans & Grefenstette (2018), the forward chaining starts from a set of initial *facts*, which are essentially the grounding of base predicates. The forward chaining process sequentially applies rules over the *fact* set, and concludes new *facts*. In NLM, we represent facts as the $\mathcal{U}$-grounding of predicates.

If we consider a set of rules that does not have recursive references, all rules can be sorted in an topological order $\mathcal{R} = (r_1, r_2, \ldots, r_k)$. We only allow references of $r_i$ from $r_j$, where $i < j$. Without loss of generality, we assume that the grounding of $r_k$ is of interest. Given the topologically resolved set of rules $\mathcal{R}$, we build a set of NLMs where each NLM realizes a specific rule $r_i$. By stacking the NLMs sequentially, we can conclude $r_k$. As a side note, for multiple rules referring to the same *head predicate* $\hat{p}$, they implicitly indicate the logical disjunction ($\vee$) of the rules. We can rename these head predicates as $\hat{p}_1, \hat{p}_2, \cdots$, and use an extra NLM to implement the logical disjunction of all $\hat{p}_i$'s.

# E  SAMPLE BLOCKS WORLD RULES

This example shows a complex reasoning in the seemingly simple Blocks World domain, which can be solved by our NLMs but requires great efforts of create manual rules by *human experts* in contrast.

Suppose we are interested in knowing *whether a block should be moved* in order to reach the target configuration. Here, a block should be moved if (1) it is moveable; and (2) there is at least one block below it that does not match the target configuration. Call the desired predicate "ShouldMove(x)".

**Input Relations.**  (Specified in the last paragraph of Section 3.4) SameWorldID, SmallerWorldID, LargerWorldID; SameID, SmallerID, LargerID; Left, SameX, Right, Below, SameY, Above. The relations are given on all pairs of objects across both worlds.

Here is one way to produce the desired predicate by defining several helper predicates, designed by "human experts":

1.  IsGround(x) $\leftarrow$ $\forall$ y Above(y, x)
2.  SameXAbove(x, y) $\leftarrow$ SameWorldID(x, y) $\wedge$ SameX(x, y) $\wedge$ Above(x, y)
3.  Clear(x) $\leftarrow$ $\forall$ y $\neg$ SameXAbove(y, x)
4.  Moveable(x) $\leftarrow$ Clear(x) $\wedge$ $\neg$ IsGround(x)
5.  InitialWorld(x) $\leftarrow$ $\forall$ y $\neg$ SmallerWorldID(y, x)
6.  Match(x, y) $\leftarrow$ $\neg$ SameWorldID(x, y) $\wedge$ SameID(x, y) $\wedge$ SameX(x, y) $\wedge$ SameY(x, y)
7.  Matched(x) $\leftarrow$ $\exists$ y Match(x, y)
8.  HaveUnmatchedBelow(x) $\leftarrow$ $\exists$ y SameXAbove(x, y) $\wedge$ $\neg$ Matched(y)
9.  ShouldMove(x) $\leftarrow$ InitialWorld(x) $\wedge$ Moveable(x) $\wedge$ HaveUnmatchedBelow(x)

We can also write the logic forms in one line: ShouldMove(x) $\leftarrow$ ($\forall$ y $\neg$ SmallerWorldID(y, x)) $\wedge$ ($\forall$ y $\neg$ (SameWorldID(y, x) $\wedge$ SameX(y, x) $\wedge$ Above(y, x))) $\wedge$ $\neg$ ($\forall$ y Above(y, x)) $\wedge$ (($\exists$ y SameWorldID(x, y) $\wedge$ SameX(x, y) $\wedge$ Above(x, y)) $\wedge$ $\neg$ ($\exists$ z $\neg$ SameWorldID(y, z) $\wedge$ SameID(y, z) $\wedge$ SameX(y, z) $\wedge$ SameY(y, z)) ).

Note that this is only a part of logic rules needed to complete the Blocks World challenge. The learner also needs to figure out where should the block be moved onto. The proposed NLM can learn policies that solve the Blocks World from the sparse reward signal indicating only whether the agent has finished the game. More importantly, the learned policy generalizes well to larger instances (consisting more blocks).

# F  IMPLEMENT NLM IN TENSORFLOW

The following python code contains a minimal implementation for one Neural Logic Machines layer with breadth equals 3 in TensorFlow. The `neural_logic_layer_breath3` is the main function. The syntax is highlighted and is best viewed in color.

```python
from itertools import permutations
import tensorflow as tf
from tensorflow.layers import dense

def expand(input, M):
  """Expands input at its second last dimension (e.g., [B, ...,
  ↪ Ni, Nj] to [B, ..., Ni, M, Nj]) by replicating tensors."""
  ndims = input.get_shape().ndims + 1
  multiples = [M if i == ndims - 2 else 1 for i in range(ndims)]
  return tf.tile(tf.expand_dims(input, -2), multiples)

def reduce(input, M):
  """Reduces max and min at the second last dimension, except for
  ↪ diagonal elements."""
  mask = _reduce_mask(input, M)[tf.newaxis, ..., tf.newaxis]
  return tf.concat([
    tf.reduce_max(input * mask, -2),
    tf.reduce_min(input * mask + (1 - mask), -2)
  ], -1)

def neural_logic(input, hidden_dim):
  """An MLP layer applied on permutations of the input."""
  return dense(_input_permutations(input), hidden_dim,
  ↪ activation=tf.sigmoid)

def neural_logic_layer_breath3(input0, input1, input2, input3, M,
↪ hidden_dim, residual):
  """A neural logic layer with breath 3.
  Args:
    input0: float Tensor of shape [B, hidden_dim], nullary
  ↪ predicates.
    input1: float Tensor of shape [B, M, hidden_dim], unary
  ↪ predicates.
    input2: float Tensor of shape [B, M, M, hidden_dim], binary
  ↪ predicates.
    input3: float Tensor of shape [B, M, M, M, hidden_dim],
  ↪ tenary predicates.
    M: int, number of objects.
    hidden_dim: int, hidden dimension.
    residual: boolean, use the residual link or not.
  Returns:
    4 float Tensors, output nullary, unary, binary tenary
  ↪ predicates respectively.
  """
  agg0 = tf.concat([input0, reduce(input1, M)], -1)
  agg1 = tf.concat([input1, expand(input0, M), reduce(input2,
    ↪ M)], -1)
  agg2 = tf.concat([input2, expand(input1, M), reduce(input3,
    ↪ M)], -1)
  agg3 = tf.concat([input3, expand(input2, M)], -1)
  outputs = [neural_logic(x, hidden_dim) for x in [agg0, agg1,
    ↪ agg2, agg3]]
  if residual:
    outputs = [tf.concat([x, y], -1) for x, y in zip(outputs,
      ↪ [input0, input1, input2, input3])]
  return outputs

def _reduce_mask(input, M):
```

```
46    dimension = input.get_shape().ndims - 2
47    base = 1.0 - tf.eye(M)
48    if dimension < 2: return tf.constant(1.0)  # Identity.
49    elif dimension == 2: return base # Diagonal excluded.
50    elif dimension == 3: return tf.expand_dims(base, 2) *
      ↪ tf.expand_dims(base, 1) * tf.expand_dims(base, 0)  # Mask
      ↪ out all tuples (x, y, z) that x == y or y == z or z == x.
51    else: raise NotImplementedError()
52
53  def _input_permutations(input):
54    dimension = input.get_shape().ndims - 2
55    if dimension < 2: return input
56    else: return tf.concat([
57      tf.transpose(input, [0] + list(perm) + [1 + dimension])
58      for perm in permutations(range(1, 1 + dimension))
59    ], -1)
```

