# OpenReview forum: "Neural Logic Machines"
_ICLR.cc/2019/Conference_

### Official Review · AnonReviewer3 · 2018-11-02
**interesting directions but unclear novelty and some claims that are too strong**

**Rating:** 5
**Confidence:** 5

**Review:**

The paper introduces Neural Logic Machines, a particular way to combine neural networks and first order but finite logic.

The paper is very well written and structured. However, there are also some downsides.

First of all, Section 2.1 is rather simple from a logical perspective and hence it is not clear what this gets a special term. Moreover, why do mix Boolean logic (propostional logic) and first order logic? Any how to you deal with the free variables, i.e., the variables that are not bounded by a quantifier? The semantics you define later actually assumes that all free variables (in your notation) are bounded by all quantifiers since you apply the same rule to all ground instances. Given that you argue that you want a neural extension of symbolic logic ("NLM is a neural realization of (symbolic) logic machines") this has to be clarified as it would not be an extension otherwise.

Furthermore, Section 2.2 argues that we can use a MLP with a sigmoid output to encode any joint distribution. This should be proven. It particular, given that the input to the network are the marginals of the ground atoms. So this is more like a conditional distribution? Moreover, it is not clear how this is different to other approaches that encode the weight of weighted logical rule (e.g. in a MLN) using neural networks, see
e.g.

Marco Lippi, Paolo Frasconi:
Prediction of protein beta-residue contacts by Markov logic networks with grounding-specific weights.
Bioinformatics 25(18): 2326-2333 (2009)

Now of course, and this is the nice part of the present paper, by stacking several of the rules, we could directly specify that we may need a certain number of latent predicates.
This is nice but it is not argued that this is highly novel. Consider again the work by Lippi and Frasconi. We unroll a given NN-parameterized MLN for s fixed number of forward chaining steps. This gives us essentially a computational graph that could also be made differentiable and hence we could also have end2end training. The major difference seems to be that now objects are directly attached with vector encodings, which are not present in Lippi and Frasconi's approach. This is nice but also follows from Rocktaeschel and Riedel's differentiable Prolog work (when combined with Lippi and Frasconi's approach).
Moreover, there have been other combinations of tensors and logic, see e.g.

Ivan Donadello, Luciano Serafini, Artur S. d'Avila Garcez:
Logic Tensor Networks for Semantic Image Interpretation.
IJCAI 2017: 1596-1602

Here you can also have vector encodings of constants. This also holds for

Robin Manhaeve, Sebastijan Dumancic, Angelika Kimmig, Thomas Demeester, Luc De Raedt:
DeepProbLog: Neural Probabilistic Logic Programming. CoRR abs/1805.10872 (2018)

The authors should really discuss this missing related work. This should also involve
a clarification of the "ILP systems do not scale" statement. At least if one views statistical relational learning methods as an extension of ILP, this is not true. Probabilistic ILP aka statistical relational learning has been used to learn models on electronic health records, see e.g., the papers collectively discussed in

Sriraam Natarajan, Kristian Kersting, Tushar Khot, Jude W. Shavlik:
Boosted Statistical Relational Learners - From Benchmarks to Data-Driven Medicine. Springer Briefs in Computer Science, Springer 2014, ISBN 978-3-319-13643-1, pp. 1-68

So the authors should either discuss SRL and its successes, separating SRL from ILP, or they cannot argue that ILP does not scale. In the related work section, they decided to view both as ILP, and, in turn, the statement that ILP does not scale is not true. Moreover, many of the learning tasks considered have been solved with ILP, too, of course in the ILP setting. Any ILP systems have been shown to scale beyond those toy domains.
This also includes the blocks world. Here relational MDP solvers can deal e.g. with BW worlds composed of 10 blocks, resulting in MDPs with several million states. And the can compute relational policies that solve e.g. the goal on(a,b) for arbitrary number of blocks. This should be incorporated in the discussion of the introduction in order to avoid the wrong impression that existing methods just work for toy examples.

Coming back to scaling, the current examples are on rather small datasets, too, namely <12 training instances. Moreover, given that we learn a continuous approximation with a limit depth of reasoning, it is also very likely that the models to not generate well to larger test instances. So the scaling issue has to be qualified to avoid to give the wrong impression that the present paper solves this issue.

Finally, the BW experiments should indicate some more information on the goal configuration. This would help to understand whether an average number of moves of 84 is good or bad. Moreover, some hints about the MDP formulation should be provided, given that there have been relational MDPs that solve many of the probabilistic planning competition tasks. And, given that the conclusions argue that NLMs can learn the "underlying logical rules", the learned rules should actually be shown.

Nevertheless, the direction is really interesting but there several downsides that have to be addressed.

---

> ### Author Response · Authors · 2018-11-17
> **Response to AnonReviewer3**
>
> We thank the reviewer for many comments and pointers, and will revise our paper to emphasize further our contributions and novelties compared to previous work.
>
> 1. Section 2.1 and the handling of free variables.
> Section 2.1 lists three primitive rules that serve as building blocks in later subsections to implement a Neural Logic Machine.  This is necessary for providing terminology and notation used throughout the rest of the paper.  We are not claiming them as novel contributions.
> Section 2.1 does *not* describe propositional logic.  The rule for “Boolean logic” is used in NLM as a component for realizing first-order logic (probabilistically, as described in section 2.2): they are used to operate on predicates grounded on objects.  An example in the Blocks World domain may look like:
>       IsGround(A) V Clear(A) -> Placeable(A)
> where A is one object in the Blocks World domain; and notably, IsGround(.), Clear(.) and Placeable(.) are not manually specified but are learned by the network.
> Our model supports free variables.  The arity of a predicate is its number of free variable.  For example, the arity of a binary predicate is 2, and NLM uses a matrix (a tensor of dimension 2) to represent the predicate’s values for all possible grounding; the 1st paragraph of section 2.2 gives further details.  The three rules (eqns 1-3) keep the same number of free variables, increase it by 1, and decrease it by 1, respectively.
>
> 2. The probability distributions modeled by MLPs.
> We would like to thank R3 for the comment about “joint distribution”, and briefly clarify technical details in Section 2.2 & 2.3 to avoid potential misunderstanding.
>
> Let’s define the input of each layer k as H_k (whose each element is in [0, 1]) recursively in the following:
>
> (1) The initial layer is H_1 = prob(B) representing boolean values 0 or 1, where B is a set of base predicates.
> (2) For each layer k, the probabilistic boolean expression in the building block is defined above Eqn. 4:
>    Expression(H_1, ... , H_k)  ==>  H_{k+1}
> where Expression in NLM is represented by some neural network structure. As illustrated in Figure 2 & 3, we use (a) grouped MLP with weights \theta_k and activation \sigma, and (b) ReduceOrExpand that computes
>    H'_k = \sigma(MLP(H_1, ... , H_k; \theta_k)),
>    H_{k+1} = ReduceOrExpand(H'_k).
> This building block keeps all elements of H_{k+1} in [0, 1] and becomes the input of next layer k+1. Therefore, such a series of building blocks is able to model a complex expression.
>
> We will not use “joint distribution” to avoid confusion, and make it more clear in the revision.
>
> 3. The difference with other approaches that encode the weight of weighted logic rules using neural networks.
> Thanks for the pointers. We will cite and discuss the papers in the revision. Our work differs substantially from MLN with weights computed by NNs, e.g., the mentioned L&F paper:
> Their logic rules (called “knowledge base” in L&F) are designed by experts; see sec 2.3 of L&F).  Here, our NLM uses deep NNs to learn such rules from data. The Blocks World example in our response to the scalability question shows the complexity of the rules that NLMs can handle.
> Consequently, our NLM needs to learn weights that form those rules. In contrast, MLN only needs to learn a real-valued weight for each hand-designed logic rule.
>
> 4. The difference with the unrolled computation graph of MLN.
> One of our main contributions is to use deep NN to learn logic rules.  Unrolling NN-parameterized MLNs is limited by the need and quality of expert-designed logic rules.
>
> 5. The encoding of objects.
> It is unclear to us what the reviewer means by “objects are … vector encodings” and hence the similarity to DeepProbLog, as we do *not* encode objects by vectors.  Data representations in NLM are all tensors that encode the (probabilistic) true/false values of grounded predicates; see the 1st paragraph of section 2.2 (page 3).

---

> > ### Author Response · Authors · 2018-11-17
> > **Response to AnonReviewer3 Continued**
> >
> > 6. The scalability discussion with ILP systems and SRL methods.
> > Thank you for the comment.  Please see our response to the scalability claim.  We will revise the paper accordingly to clarify.
> >
> > 7. Generalization w.r.t. the number of objects.
> > Different from the reviewer’s hypothesis, our results actually verify that NLM models do generalize well to larger test instances.  For example, Table 2 shows that our learned model achieves 100% accuracy on test instances with more blocks, and the same for Table 1. We have also conducted experiments testing this ability using several trained model in extreme cases which consist of 500 blocks (1000 numbers for sorting), no failure cases were found. The models will be made public along with our code after the paper decision. This ability is one of our main findings, as highlighted in the abstract (“NLMs ... generalize to arbitrarily large-scale tasks”).
> >
> > 8. The goal configuration of Blocks World.
> > We present the generation of Blocks World instances in Appendix B.4. We will make it more clear in the revision. The goal configuration is randomly and independently generated as the initial configuration. One can compute that the expected optimal number of steps needed for solving the Blocks World is approximately 2m - o(m) steps are required to solve the case, where m is the number of blocks, which is 50 in the test instances. In average 84 steps means the model learns a fairly good solution. The reviewer is also welcome to check our demo in the footnote of Page 8: https://sites.google.com/view/neural-logic-machines .
> >
> > 9. MDP formulation of the Blocks World.
> > Thanks for the nice suggestion. We discuss the MDP formulation in section 3.4, and we will make it more clear. We input the current world and the target world with tensors describing relations between objects. At each time step, the agents take actions to move one block onto another. We use sparse rewards to train the agents: The agents get the reward only when they finish the task.
> >
> > 10. NLM learns the underlying logical rules.
> > Thanks for the comment.  We intend to mean that the learned NLM generalizes well to problems with varying sizes, in the same way logical rules do.  We will reword the sentence to avoid confusions, and discuss rule extraction as future work.

---

### Official Review · AnonReviewer2 · 2018-11-02

**Rating:** 7
**Confidence:** 2

**Review:**

In this paper the authors propose a neural-symbolic architecture, called Neural Logic Machines (NLMs), that can learn logic rules.

The paper is pretty clear and well-written and the proposed system is compelling. I have only some small concerns.
One issue concerns the learning time. In the experimental phase the authors do not state how long training is for different datasets.
Moreover it seems that the “rules” learnt by NSMs cannot be expressed in a logical formalism, isn’t it? If I am right, I think this is a major difference between dILP (Evans et. al) and NLMs and the authors should discuss about that. If I am wrong, I think the authors should describe how to extract rules from NLMs.
In conclusion I think that, once these little issues are fixed, the paper could be considered for acceptance.

[minor comments]
p. 4
“tenary” -> “ternary”
 p. 5
“ov varying size” -> “of varying size”
“The number of parameters in the block described above is…”. It is not clear to me how the number of parameters is computed.
“In Eq. equation 4” -> “In Eq. 4”

p. 16
“Each lesson contains the example with same number of objects in our experiments.”. This sentence sounds odd.

---

> ### Author Response · Authors · 2018-11-21
> **Response to AnonReviewer2**
>
> 1. Running time / training time.
> The number of examples/episodes used is shown in Table 4. We plan to add training time / inference speed in our revision. Here, we show our results on Blocks World. We train our model on 12 CPUs (Xeon E5) and a single GPU (GTX 1080), It takes 3 hours to train our model (26000 episodes). During inference, the model runs in 1.43s per episode when the number of blocks is 50.
>
> 2. Rules are not expressed in a logical formalism.
> Thanks for the comment and suggestion --- Yes, your understanding is correct. Although the design of NLM’s neural architecture is highly motivated by FOPC logic formalism, NLM models do not explicitly encode FOPC logic forms. In contrast, the weights of the MLPs encodes how models should perform the deduction, and the output of the NLM can be regarded as the conclusions (0/1 indicating whether we should move the block, in a Blocks World).

---

### Official Review · AnonReviewer1 · 2018-11-05
**Interesting approach to model FOL in NN, with concerns in scalability**

**Rating:** 6
**Confidence:** 3

**Review:**

This paper presents a model to combine neural network and logic programming. It proposes to use 3 primitive logic rules to model first-order predicate calculus in the neural networks. Specifically, relations with different numbers of arguments over all permutations of the groups of objects are represented as tensors with corresponding dimensions. In each layer, a MLP (shared among different permutations) is applied to transform the tensor. Multiple layers captures multiple steps of deduction. On several synthetic tasks, the proposed method is shown to outperform the memory network baseline and shows strong generalization.

The paper is well written, but some of the contents are still a bit dense, especially for readers who are not familiar with first-order predicate calculus.

The small Python example in the Appendix helps to clarify the details. It would be good to include the details of the architectures, for example, the number of layers, and the number of hidden sizes in each layer, in the experiment details in the appendix.

The idea of using the 3 primitive logic rules and applying the same MLP to all the permutations are interesting. However, due to the permutation step, my concern is whether it can scale to real-world problems with a large number of entities and different types of relations, for example, a real-world knowledge graph.

Specifically:

1. Each step of the reasoning (one layer) is applied to all the permutations for each predicate over each group of objects, which might be prohibitive in real-world scenario. For example, although there are usually only binary relations in real-world KG, the number of entities is usually >10M.

2. Although the inputs or preconditions could be sparse, thus efficient to store and process, the intermediate representations are dense due to the probabilistic view, which makes the (soft) deduction computationally expensive.

Some clarification questions:

Is there some references for the Remark on page 3?

Why is there a permutation before MLP? I thought the [m, m-1, …, m-n+1] dimensions represent the permutations. For example, if there are two objects, {x1, x2}. Then the [0, 1, 0] represents the first predicate applied on x1, and x2. [1, 0, 0] represents the first predicate applied on x2 and x1. Some clarifications would definitely help here.

I think this paper presents an interesting approach to model FOPC in neural networks. So I support the acceptance of the paper. However, I am concerned with its scalability beyond the toy datasets.

---

> ### Author Response · Authors · 2018-11-21
> **Response to AnonReviewer1**
>
> 1. Model details.
> Detailed implementation details including the number of layers (a.k.a. the depth) can be found in Table 4 (Appendix B.2). As for the hyper-parameters of the MLPs, we use no hidden layer, and the hidden dimension (number of intermediate predicates) of each layer is set to 8 across all our experiments.
> We thank the reviewer for the suggestion, and will make these information more clear in our revision. Moreover, we plan to release our code upon acceptance.
>
> 2. Scalability
> It should be clarified that scalability mentioned in the paper mainly refers to the complexity of reasoning (e.g., number of steps before producing a desired predicate), not the number of objects/entities or relations. For example, as shown in our general clarification, learning predicates that have a complex structure (such as the ShouldMove in the example) pose a scalability challenge to existing ILP methods. We also refer the reviewer to our clarification on scalability for more detailed analysis.
> In general, we agree with the reviewer that an inductive logic system should be able to handle both complex reasoning rules (e.g., as the settings explored in our paper) and large-scale entity sets (e.g., as in knowledge graph-related literature). We hope the methods and insights we presented in this paper could help the whole community in this interesting direction.
>
> 3. Permutation in MLP.
> Permutation is needed in two places.  Consider two n-ary MLPs at a particular layer of the NLM (called “p”).  As the reviewer correctly points out, the [m, m-1, …, m-n+1] dimensions represent permutations in the input of p.   On the other hand, the permutation before MLP is to create new predicates that only differs from the existing one in the variable order, in order to compute composition of these two predicates; this is the second place where permutation is needed.
>
> As an example, suppose “p” is the predicate HasEdge(x, y).  By permuting its variables, we get another predicate, HasReverseEdge(x, y), which is TRUE if there is an edge from y to x.  These two predicates can be used to compose a more complex predicate
>     HasBidirectionalEdge(x, y) ← HasEdge(x, y) ∧ HasReverseEdge(x, y)

---

> > ### Comment · AnonReviewer1 · 2018-11-27
> > **Reply to authors' response**
> >
> > Thanks for the clarification about the details and the scalability. I would like to keep my rating. This is an interesting direction and worth pursuing, so I support acceptance. But it is still unclear to me how the proposed approach can move beyond toy datasets.

---

### Author Response · Authors · 2018-11-17
**Clarification on Scalability**

We thank all reviewers for their thoughts and comments. In addition to the specific responses below, here we clarify on the scalability question asked by some reviewers. We will include related discussions in our revision.

It should be clarified that scalability mentioned in the paper mainly refers to the complexity of reasoning (e.g., number of steps before producing a desired predicate), not the number of objects/entities or relations. This is highlighted in #2 at the bottom of page 1: “We expect the learning system to scale with the number of logic rules. Existing logic-based algorithms like ILP suffer an exponential computational complexity with respect to the number of logic rules”.

Knowledge-graph tasks involve many entities (e.g. > 10M) and relations as reviewers pointed out, but the rules involved in the reasoning steps are usually restricted. For example, the rules considered in the knowledge base reasoning work (Yang et al., 2017) are restricted in a “chain-like” form (their eqn 1.), which is query(Y,X)<-Rn (Y,Zn) ∧ · · · ∧ R1 (Z1,X), while R1, . . . , Rn are *known* relations in the knowledge base. Such knowledge-graph reasoning tasks represent an interesting yet different class of problems outside of the current scope of our paper.

In contrast, learning predicates that have a complex structure (such as the ShouldMove example below) pose a scalability challenge to existing ILP methods.  In dILP [Evans et al.], for example, suppose each rule has C possible choices from the templates and R rules are need to be learned, then the possible space is at least O(C^R) --- the number of the set of possible rules is exponential w.r.t. the number of rules.  On the other hand, our method is only quadratic in the number of rules (or in this case, equivalently, number of predicates).

**********************************************************************
                          A Blocks World Example
**********************************************************************
This example shows what we mean by complex reasoning in the seemingly simple Blocks World domain.  Suppose we are interested in knowing whether a block should be moved in order to reach the target configuration.  Here, a block should be moved if (1) it is moveable; and (2) there is at least one block below it that does not match the target configuration.  Call the desired predicate “ShouldMove(x)”.

Inputs Relations (as specified in the last paragraph of page 7):
SameWorldID, SmallerWorldID, LargerWorldID;
SameID, SmallerID, LargerID;
Left, SameX, Right, Below, SameY, Above.
The relations are given on all pairs of objects across both worlds.

Here is one way to produce the desired predicate by defining several helper predicates, designed by “human experts”:
1. IsGround(x) ← ∀y Above(y, x)
2. SameXAbove(x, y) ← SameWorldID(x, y) ∧ SameX(x, y) ∧ Above(x, y)
3. Clear(x) ← ∀y ¬SameXAbove(y, x)
4. Moveable(x) ← Clear(x) ∧ ¬IsGround(x)
5. InitialWorld(x) ← ∀y ¬SmallerWorldID(y, x)
6. Match(x, y) ← ¬SameWorldID(x, y) ∧ SameID(x, y) ∧ SameX(x, y) ∧ SameY(x, y)
7. Matched(x) ← ∃y Match(x, y)
8. HaveUnmatchedBelow(x) ← ∃y SameXAbove(x, y) ∧ ¬Matched(y)
9. ShouldMove(x) ← InitialWorld(x) ∧ Moveable(x) ∧ HaveUnmatchedBelow(x)
We can also write the logic forms in one line:
ShouldMove(x) ← (∀y ¬SmallerWorldID(y, x)) ∧ (∀y ¬(SameWorldID(y, x) ∧ SameX(y, x) ∧ Above(y, x))) ∧ ¬(∀y Above(y, x)) ∧ ((∃y SameWorldID(x, y) ∧ SameX(x, y) ∧ Above(x, y)) ∧ ¬(∃z ¬SameWorldID(y, z) ∧ SameID(y, z) ∧ SameX(y, z) ∧ SameY(y, z)) )

Note that this is only a part of the logic needed to complete the Blocks World challenge. The learner also needs to figure out where should the block be moved onto. The proposed NLM can learn policies that solve the Blocks World from the sparse reward signal indicating only whether the agent has finished the game. More importantly, the learned policy generalizes well to larger instances (consisting more blocks).
**********************************************************************

---

### Public Comment · (anonymous) · 2018-12-07
**Potential related work**

... although it is not a differentiable model or even a neural model, the idea of learning to sort infinite arrays from short examples has been explored in the "Generalized Planning" literature, for example,
http://rbr.cs.umass.edu/shlomo/papers/SIZaij11.pdf
https://www.ijcai.org/Proceedings/11/Papers/159.pdf
https://www.dtic.upf.edu/~jonsson/ker18.pdf

---

> ### Author Response · Authors · 2018-12-09
> **Thanks for your pointers**
>
> Thanks for your pointers to the related papers. We will discuss them in the next version of our paper.

---

### Meta-Review · Area_Chair1 · 2018-11-06
**Interesting forward chaining approach to neural deduction**

**Confidence:** 5
**Recommendation:** Accept (Poster)

**Metareview:**


pros:
- The paper presents an interesting forward chaining model which makes use of meta-level expansions and reductions on predicate arguments in a neat way to reduce complexity.  As Reviewer 3 points out, there are a number of other papers from the neuro-symbolic community that learn relations (logic tensor networks is one good reference there). However using these meta-rules you can mix predicates of different arities in a principled way in the construction of the rules, which is something I haven't seen.
- The paper is reasonably well written (see cons for specific issues)
- There is quite a broad evaluation across a number of different tasks.  I appreciated that you integrated this into an RL setting for tasks like blocks world.
- The results are good on small datasets and generalize well

cons:
- (scalability) As both Reviewers 1 and 3 point out, there are scalability issues as a function of the predicate arity in computing the set of permutations for the output predicate computation.
- (interpretability) As Reviewer 2 notes, unlike del-ILP, it is not obvious how symbolic rules can be extracted.  This is an important point to address up front in the text.
- (clarity) The paper is confusing or ambiguous in places:

-Initially I read the 1,2,3 sequence at the top of 3 to be a deduction (and was confused) rather than three applications of the meta-rules.  Maybe instead of calling that section "primitive logic rules" you can call them "logical meta-rules".

-Another confusion, also mentioned by reviewer 3 is that you are assuming that free variables (e.g. the "x" in the expression "Clear(x)") are implicitly considered universally quantified in your examples but you don't say this anywhere.  If I have the fact "Clear(x)" as an input fact, then presumably you will interpret this as "for all x Clear(x)" and provide an input tensor to the first layer which will have all 1.0's along the "Clear" relation dimension, right?

-It seems that you are making the assumption that you will never need to apply a predicate to the same object in multiple arguments?  If not, I don't see why you say that the shape of the tensor will be m x (m-1) instead of m^2.  You need to be able to do this to get reflexivity for example: "a <= a".

-I think you are implicitly making the closed world assumption (CWA) and should say so.

-On pg. 4 you say "The facts are tensors that encode relations among multiple objectives, as described in Sec. 2.2.".  What do you mean by "objectives"?  I would say the facts are tensors that encode relations among multiple objects.

-On pg. 5 you say "We finish this subsection, continuing with the blocks world to illustrate the forward
propagation in NLM".  I see no mention of blocks world in this paragraph. It just seems like a description of what happens at one block, generically.

-In many places you say that this model can compute deduction on first-order predicate calculus (FOPC) but it seems to me that you are limited to horn logic (rule logic) in which there is at most one positive literal per clause (i.e. rules of the form: b1 AND b2 AND ... AND bn => h).  From what I can tell you cannot handle deduction on clauses such as b1 AND b2 => h1 or (h2 and h3).

-There is not enough description of the exact setup for each experiment. For example in blocks world, how do you choose predicates for each layer?  How many exactly for each experiment?  You make it seem on p3 that you can handle recursive predicates but this seems to not have been worked out completely in the appendix.  You should make this clear.

-In figure 1 you list Move as if its a predicate like On but it's a very different thing. On is  predicate describing a relation in one state.  Move is an action which updates a state by changing the values of predicates.  They should not be presented in the same way.

-You use "min" and "max" for "and" and "or" respectively.  Other approaches have found that using the product t-norm t-norm(x,y) = x * y helps with gradient propagation.  del-ILP discusses this in more detail on p 19.  Did you try these variations?

-I think it would be helpful to somewhere explicitly describe the actual MLP model you use for deduction including layer sizes and activation functions.

-p. 5. typo: "Such a parameter sharing mechanism is crucial to the generalization ability of NLM to
problems ov varying sizes." ("ov" -> "of")

-p. 6. sec 3.1 typo: "For ∂ILP, the set of pre-conditions of the symbols is used direclty as input of the system." ("direclty" -> "directly")

I think this is a valuable contribution and novel in the particulars of the architecture (eg. expand/reduce) and am recommending acceptance.  But I would like to see a real effort made to sharpen the writing and make the exposition crystal clear.  Please in particular pay attention to Reviewer 3's comments.

---

> ### Author Response · Authors · 2018-12-21
> **Thanks for your comprehensive comments. We will revise the paper accordingly.**
>
> Dear AC,
>
> Thanks for your careful reading of our manuscript and consideration.  We appreciate your remarkably comprehensive comments and suggestions. We promise to provide an inclusive revision in the camera-ready version accordingly.
>
> Many thanks,
> Authors.